# Quality of nutrition services in primary health care facilities of Dhaka city: State of nutrition mainstreaming in urban Bangladesh

**Faugia Islam Anne[1], Syeda Mahsina Akter[2], Sifat Parveen Sheikh[1], Santhia Ireen[3], Jessica Escobar-DeMarco[3], Kristen Kappos[3], Deborah Ash[3], Sabrina Rasheed[1]***

**1** Health Systems and Population Studies Division (HSPSD), Urban Health, icddr,b, Dhaka, Bangladesh, **2** Wageningen University, Wageningen, Netherlands, **3** Alive & Thrive, FHI Solutions / FHI 360, Dhaka, Bangladesh

* sabrina1@icddrb.org

## Abstract

### Introduction

Despite high prevalence of malnutrition little is known about the quality of nutrition services provided through urban health systems. This study aimed to fill in knowledge gaps on quality of nutrition service provision at public primary health care facilities in urban Dhaka.

### Method

This cross-sectional study was conducted from April-July 2019 in Dhaka City. Fifty-three health facilities were sampled following NetCode protocol. Quality of nutrition services was assessed in terms of structural readiness, process, and client satisfaction. Structural readiness included equipment, guidelines, and registers, and knowledge of health professionals (n = 130). For process, client provider interaction was observed (ANC: n = 159, Pediatric: n = 150). For outcome assessment, client's satisfaction with nutrition service provision was measured through interviews with pregnant women (n = 165) and caregivers of 0–24 month-old children (n = 162). Bivariate and multivariate analyses were conducted using SPSS.

### Results

There were gaps in availability of equipment and guidelines in health facilities. Only 30% of healthcare providers received basic nutrition training. The mean knowledge score was 5.8 (range 0–10) among ANC providers and 7.8 for pediatric service providers. Process: Only 17.6% health facilities had dedicated space for counselling, 48.4% of pregnant women received four key nutrition services; 22.6% of children had adequate growth monitoring; and 38.7% of caregivers received counselling on exclusive breastfeeding. Outcome: The mean satisfaction with services was 4.3 for ANC and 4.0 for paediatric visits (range 1–5). Participants attending public facilities had significantly lower satisfaction compared to those attending private and NGO health facilities.

**Data Availability Statement:** A sample deidentified data set is attached as a Supporting Information file. The additional data we used in this manuscript

are available from the repository of icddrb upon reasonable request by researchers via Armana Ahmed (armana@icddrb.org), Head of Research Administration of icddr,b, as per the institutional data access policy of icddr,b.

**Funding:** The funding for the study was awarded by The Bill & Melinda Gates Foundation through Alive & Thrive, managed by FHI Solutions. The funders had no role in study design, data collection and analysis, decision to publish, or preparation of the manuscript.

**Competing interests:** The authors have declared that no competing interests exist.

## Conclusion

There were gaps in facility readiness, and implementation of nutrition services. The clients were more satisfied with services at private facilities compared to public. The gaps in nutrition service delivery need to be adequately addressed to ensure promotion of good nutrition and early detection and management of malnutrition among pregnant women and children in urban Bangladesh.

## Introduction

The urban population is on the rise around the world, from 751 million in 1950 to 4.2 billion in 2018 [1]. Many developing countries including Bangladesh are undergoing rapid urbanization [2] fueled by migration from rural areas of people looking for economic opportunities. Therefore, population of informal settlements in the cities has grown at a rate of almost 7% per year [3]. Currently around 7 million people living in the informal settlements of Dhaka face challenges of overcrowding, poor infrastructure, inadequate water supply, limited access to sanitation services, and lack of access to healthcare [4, 5]. Among the residents of urban informal settlements infant mortality is higher (70/1000 live births) compared to both urban non-poor (34/1000 live births) and rural residents (40/1000 live births) indicating poor health status [6]. In terms of health care utilization, residents of urban informal settlements suffer from inequity. Among pregnant women, the rates of utilization of at least 4 antenatal care services (ANC) was 36% among those living in urban informal settlements compared to 58.7% among other urban residents [7]. Similar gaps in immunization rates are also observed for young children [8].

To address the high levels of malnutrition that exist in Bangladesh, nutrition has been mainstreamed into maternal, neonatal and child health (MNCH) services from 2010. Since its inception, National Nutrition Services (NNS) has been responsible for delivering nutrition specific services nationally through primary health care (PHC) [9]. However, the Ministry of Health and Family Welfare (MoHFW) is responsible for PHC in rural areas [10] while in urban areas it is the responsibility of Ministry of Local Government, Rural Development and Cooperatives (MoLGRD&Co, or MoLG) [11]. The MoLG contracts out the delivery of urban PHC to non-governmental organizations (NGOs) [12]. In urban areas in addition to public sector, both private and NGO sectors are involved in delivering PHC and therefore nutrition services [13]. In previous studies, urban PHC services provided were deemed to be inadequate to meet the needs of the burgeoning urban population [13, 14]. In terms of nutrition services provided through PHC, researchers have shown there are important gaps in the quality of nutrition service delivered through the public PHCs in rural areas [15] although much less is known about the quality of nutrition service provision in urban areas.

To understand the quality of nutrition services provided in urban Bangladesh, it is important to look at service provision at public, NGO and private sector as the public sector is not the main PHC service provider. Previous research demonstrated that the urban poor tend to seek health care from private healthcare facilities, pharmacies and informal healthcare providers although the services are very limited and the quality of care obtained is sub-optimal [16, 17]. Hence, the current study was conducted to identify gaps in nutrition service delivery in the urban context, which would inform future interventions.

## Methods

### Study design, setting and sample selection

We conducted a cross-sectional study from April-July 2019 in Dhaka city. To generate a list of available health facilities in Dhaka city, we used the Urban Health Atlas [18] and a list provided

**Table 1. Sample size for health facility and study participants.**

| Type of health facilities | Public | Private | NGO | Total |
|---|---|---|---|---|
| Primary health facilities | 11 | 11 | 11 | 33 |
| Secondary health facilities (delivery services) | 4 | 3 | 3 | 10 |
| Secondary health facilities (pediatric services) | 4 | 3 | 3 | 10 |
| Total health facilities | | | | 53 |
| Study Participants | | | | |
| Health facility managers | | | | 53 |
| Healthcare providers | | | | 130 |
| Pregnant women | | | | 324 |
| Mothers/ caregivers of young children | | | | 312 |

by the Directorate General of Health Services (DGHS). From the list we identified health facilities that provided ANC and postnatal care (PNC), delivery services, and pediatric care (immunization and other outpatient and inpatients services for children). The list was verified through field visits and information was collected on service utilization for the previous month. Based on service utilization rates, we identified facilities that provided services to at least 15 mothers or young children per day. The identified facilities were stratified according to the types of service providers (public n = 18, NGO n = 19, and private n = 42). We followed the WHO/UNICEF NetCode protocol for periodic facility assessment [19]. Although NetCode recommended the selection of 33 primary healthcare facilities in total using Probability Proportional to Size (PPS) sampling, we did not consider facility size in the selection of facilities and randomly selected equal number of facilities from public, NGO, and private sector with equal probabilities of selection for our study. For secondary-level health facilities, we listed facilities that provide normal delivery (maternal) and/or in-patient services for common childhood illnesses (pediatric). From the list, we identified those with the highest patient flow and selected 10 facilities each that provided maternal and pediatric services (Table 1).

## Study participants and sample size

We interviewed 53 health facility managers and 130 healthcare providers (doctors, nurses, midwives, health workers) (Table 1). From each primary health facility, we selected 5 pregnant women and 5 mothers/ caregivers of young children each for exit interview and observation in first come first serve basis. The final sample included pregnant women (observation n = 159, exit interview n = 165) and mothers/ caregivers of young children (observation n = 150, exit interview n = 162). In a few cases, the service providers did not allow us to observe service provision and other cases mothers did not consent to be interviewed. If one respondent did not consent, we approached the next mother or caregiver.

## Data collection

We invited the health facility owners or managers to a meeting organized by DGHS to sensitize them about our study. Upon obtaining their consents, we contacted them for an appointment to conduct our study with a support letter from DGHS. The interviews with health facility managers and healthcare providers were conducted at a time and place convenient for the respondents. At the primary healthcare facilities, we visited the outdoor service provision areas during the work hours (9 am- 1 pm and 4 pm—8 pm) for observation and exit interviews. Permission was obtained from healthcare providers and their clients for data collection. Data collection in each facility was completed within 1–2 days.

The study tools used for data collection were adapted and validated for Bangladesh [15]. The tools were translated into Bengali, pre-tested and slightly modified for clarity. The data collection team received 4-day training on the tools and data collection techniques. Data were collected on paper and a digital data entry form was used to enter the data. The data were reviewed for consistency and quality and finalized for analysis.

## Assessment of quality of care

Quality of care was assessed based on three dimensions: structural readiness, process and outcome [20], each of which had multiple indicators and data collection tools (Table 2).

The indicators for structural readiness of the health facility included availability of equipment, guidelines supplements [vitamin A, Iron, Folic Acid (IFA), Calcium], and registers/ reporting forms. In addition, we collected information on training and knowledge of the healthcare providers on nutrition. The indicators for process included nutrition service provision during ANC and pediatric visits in the PHC facilities only. The indicators for outcome of care included satisfaction of pregnant women and mothers/ caregivers with the nutrition service provided through PHC.

## Development of composite indicators

**Knowledge score.** Healthcare providers were asked 10 questions each on nutrition topics to address during ANC and pediatric visits. For each correct answer, a score of 1 was given while incorrect answers were given a score 0. The total score was created by adding the scores obtained from the questions of the scale. Range of knowledge score was between 0–20 [15].

**Satisfaction score.** Clients of nutrition services were asked 7 questions related to satisfaction with the services received. The questions were related to cleanliness of waiting area, waiting time, quality of advice provided, comfort about asking questions, respectful treatment, and intention to use the services again or recommend the services to others. For each question, the response was recorded on a five-point Likert scale with the score of 1 being 'very poor' and 5 being 'excellent'. For each respondent, the scores from each of 7 questions were added to create a total score. The total score was divided by 7 to obtain mean satisfaction score [21].

## Data analysis

For structural readiness, we used frequency distributions to assess readiness of the heath facility to deliver nutrition services. To assess the healthcare providers' knowledge, we assessed the difference in mean knowledge scores between ANC and pediatric service providers using t-test statistics; analysis was adjusted for clustering and sampling weights to ensure

**Table 2. Indicators and tools for assessment of the quality of nutrition services.**

| Dimensions | Indicator/information | Tools |
|---|---|---|
| Structural readiness | Availability of <br> • equipment <br> • supplements vit. A, IFA, calcium <br> • guidelines <br> • registers | Record review and observation of equipment and job aids |
| | | Interview with health facility manager |
| | • nutrition training and knowledge of health care provider | Interview with health care providers |
| Process | Nutrition service delivery | Observation of client-service provider interaction |
| Outcome | Pregnant women/ caregiver's satisfaction of services | Exit interview |

representativeness. The association between the process of nutrition services and characteristics of care recipients was tested using chi-square test adjusted for clustering and sampling weights. For satisfaction scores, Cronbach's alpha (which was 0.63 and considered acceptable) was used to test the internal consistency and reliability of the questionnaire for each domain [22]. Multivariable analyses were employed to identify factors associated with overall satisfaction score among respondents, analysis was adjusted for clustering and sampling weights to ensure representation. Data were analyzed using SPSS version 25.

### Ethical clearance

The Ethical Review Committee (ERC) of the icddr,b approved the study (Protocol Number # 18092). The Institutional Review Board of FHI 360 acknowledged the study as expedited (Project # 1368307–2). Written informed consent was obtained from participants after explaining study objectives and possible benefits and risks associated with the study. Privacy and confidentiality of the facility and respondents were strictly maintained.

## Results

All primary health facilities provided services such as ANC, PNC, immunization and management of childhood illness. Maternity facilities provided both inpatient and outpatient services and focused on pregnancy and childbirth. Pediatric facilities had both inpatient and outpatient services and focused on providing healthcare for infants and young children. Secondary health facilities (maternity and pediatric) employed specialists (gynecologists and obstetricians, and pediatricians) in addition to general practitioners/medical officers and nurses, while primary health facilities employed general practitioners/medical officers, nurses and paramedics.

### Structural readiness of the health facilities

In terms of equipment, most facilities had adult weighing scales (96.2%), stadiometers (96.2%), and registers (90.5%). Social and behavior change communication (SBCC) materials on maternal nutrition were less available (69.8%). About half of the facilities had calcium supplements, less than one-third of the facilities had IFA supplements, and 64% health facilities had vitamin A supplements (post-partum). Less than half of facilities had basic training guidelines on nutrition and about a third of facilities had guidelines for IFA and calcium distribution (Table 3). In terms of equipment for pediatric services, most facilities had infant weighing scales. However, some gaps remained around availability of child weighing scales, length boards, mid-upper arm circumference (MUAC) tapes, growth monitoring cards and SBCC materials on Infant and Young Child Feeding (IYCF) (Table 3).

Among the healthcare providers interviewed in the primary healthcare facilities, 60% were doctors, followed by nurses and paramedics. The healthcare providers were predominantly female and only a third of the providers (31.3% of ANC service providers and 39.7% pediatric service providers) received basic training on nutrition provided by NNS (Table 4). It is important to note that the service providers reported receiving nutrition training (especially maternal nutrition) from sources other than NNS.

In terms of knowledge about topics of nutrition counseling to be included during ANC, the mean knowledge score obtained by the primary healthcare providers was 5.8 (out of 10). 90% of ANC providers knew that pregnant women need to be counselled about exclusive breastfeeding for 6 months. More than 80% of ANC service providers knew that pregnant women need to be counselled about consuming animal source foods, fruits and vegetables, and calcium supplements. However, less than 50% of ANC providers knew that they had to provide information to mothers about iron folic acid supplementation and early initiation of

**Table 3. Availability of equipment, supplements, guidelines and registers.**

| | Primary healthcare facilities | Maternity health facilities | Pediatric health facilities |
|---|---|---|---|
| | N = 33 | N = 10 | N = 10 |
| *Antenatal/Postnatal care service* | 33 | 10 | 8 |
| Weighing scale | 33 | 10 | 8 |
| Stadiometer | 29 | 8 | 7 |
| Maternal body weight monitoring cards | 22 | 7 | 6 |
| SBCC materials (maternal nutrition) | 25 | 8 | 4 |
| Calcium supplement | 14 | 7 | 5 |
| IFA[a] supplement | 8 | 2 | 5 |
| Vitamin A (post-partum) | 22 | 7 | 5 |
| Register/reporting forms | 32 | 8 | 8 |
| Baby-Friendly Hospital Initiative (BFHI) guidelines | 9 | 6 | 4 |
| Basic training guidelines on nutrition | 16 | 5 | 4 |
| Guidelines for distribution of IFA | 10 | 3 | 1 |
| Guidelines for distribution of calcium | 11 | 3 | 2 |
| Facilities with >5 items of equipment | 33 | 9 | 8 |
| Facilities with none of the equipment | 0 | 0 | 0 |
| *Pediatric/ immunization service* | 33 | 10 | 10 |
| Infant weighing scale | 32 | 10 | 10 |
| Child weighing scale | 26 | 6 | 7 |
| Height/length board | 27 | 9 | 8 |
| Tape measures (MUAC) | 26 | 6 | 9 |
| Growth charts | 22 | 5 | 6 |
| IMCI[b] chart booklet | 18 | 3 | 5 |
| SBCC materials on IYCF[c] | 31 | 9 | 9 |
| Register/reporting forms | 28 | 7 | 8 |
| IYCF Strategy [2007] | 12 | 3 | 5 |
| IYCF manual [2011–12] | 8 | 3 | 4 |
| Distribution of Vitamin-A | 12 | 5 | 5 |
| Facilities with >5 items of equipment | 32 | 10 | 10 |
| Facilities with none of the equipment | 0 | 0 | 0 |

[a]IFA = Iron folic acid

[b]IMCI = Integrated management of childhood illness

[c]IYCF = Infant and young child feeding

breastfeeding. Only 2% of ANC providers knew that they had to provide information to pregnant mothers on the dangers of introducing liquids early (Table 5).

In terms of knowledge about topics of nutrition counselling on IYCF and feeding during illness among pediatric service providers, the mean knowledge score was 7.8 (out of 10). More than 90% of pediatric service providers knew that they need to counsel mothers about exclusive breastfeeding for 6 months, continuing breastfeeding in case of illness of mother (child <6m old), and providing zinc during a child's diarrhoeal episode (child >6m old) (Table 5). However, less than half of the pediatric service providers knew that they need to counsel mothers about the timing of introduction of water or other liquids, semi-solid and animal source foods and breastfeeding more frequently if the baby is not getting enough milk. There was no significant difference observed between ANC service providers and pediatric service providers in terms of knowledge scores (data not shown).

**Table 4. Characteristics of primary healthcare providers.**

| Characteristics of primary healthcare providers | ANC[a] service providers | Pediatric service providers |
|---|---|---|
| | n = 80 | n = 63 |
| | % (Mean ± SD) | % (Mean ± SD) |
| *Age*(years) | (37.9 ± 10.9) | (41.6 ± 12.3) |
| *Sex* | | |
| Male | 1.3 | 25.4 |
| Female | 98.8 | 74.6 |
| *Years of schooling* | (17.0 ± 2.8) | (17.2 ± 3.0) |
| *Type of health care providers* | | |
| Doctors | 60 | 60 |
| Nurse | 21.3 | 9.5 |
| Midwife | 1.3 | 0 |
| Paramedic | 15 | 22 |
| Nutritionist | 0 | 1.6 |
| *Nutrition training* | | |
| Basic nutrition (NNS) | 31.3 | 39.7 |
| IYCF (NNS) | 26.3 | 27.0 |
| Growth monitoring (NNS) | 27.5 | 31.7 |
| Severe Acute Malnutrition management (NNS) | 23.8 | 42.9 |
| Maternal nutrition training | 20 | 20.6 |
| Child nutrition training | 13.8 | 17.5 |
| Any other training | 1.3 | 3.2 |

[a] ANC = Antenatal care

NNS = National Nutrition Services

### Process of nutrition service delivery

During ANC visits, we found that most mothers were weighed (89.9%), given dietary advice (84.2%) and provided with IFA (69.8%) and calcium supplements (84.2%). However, only 11% of the mothers were counseled on breastfeeding and 18% of health facilities had dedicated space for counselling (Fig 1A). During pediatric visits, most children were weighed (79.3%) and their feeding practice was assessed (84%). However, in only 22.6% of observations, the child's height was measured, and weight assessed against the growth chart. There were gaps in the use of SBCC materials, and counseling on exclusive breastfeeding, feeding frequency and Vitamin A supplementation (Fig 1B). There was no significant association observed between the process of nutrition services and characteristics of care recipients (data not shown).

### Outcome: Patient's satisfaction

In terms of satisfaction with nutrition services, the mean satisfaction score for pregnant women was 4.1 and for mothers /caregivers of young children were 4.0 (range 1–5), respectively, indicating that the women were fairly satisfied with the services provided. For both pregnant women and mothers/ caregivers of young children, mean satisfaction scores were the lowest for the domains "waiting time" and "waiting area" (Table 6).

Multivariate analysis revealed that the type of health facility, number of nutrition services received and the age of the pregnant woman were significantly associated with satisfaction (Table 7). Pregnant women aged below 30 years reported significantly higher satisfaction compared to those aged 30 years and above; pregnant women attending public health facilities

**Table 5. Healthcare provider's knowledge on nutrition topics for counselling pregnant and lactating mothers.**

| Nutrition topics | ANC providers n = 80 | Paediatric service providers n = 63 | All providers n = 130 |
|---|---|---|---|
| | % (Mean ± SD) | % (Mean ± SD) | % (Mean ± SD) |
| *During ANC mothers need to be advised on* | | | |
| Animal source food | 85 | 71.4 | 79.2 |
| Vitamin A rich foods | 71.3 | 66.7 | 68.5 |
| Fruits and vegetables | 82.5 | 77.8 | 59.2 |
| Frequency of food consumption | 62.5 | 61.9 | 83.8 |
| Iodized Salt | 11.3 | 4.8 | 10 |
| Iron folic acid supplementation | 46.3 | 57.1 | 50.8 |
| Calcium supplementation | 87.5 | 84.1 | 85.4 |
| Early initiation of breastfeeding | 48.8 | 60.3 | 54.6 |
| Exclusive breastfeeding for 6 m | 90 | 79.4 | 84.6 |
| Dangers of introducing liquids early | 2 | 0 | 0.8 |
| Knowledge score (scale 0 to 10) | (5.8 ± 1.5) | (5.6 ± 1.5) | (5.7 ± 1.5) |
| *IYCF[a] and feeding during illness* | | | |
| Breastfeeding initiation within 1hour of birth | 62.5 | 73 | 66.9 |
| Exclusive breastfeeding for 6m | 100 | 98.4 | 99.2 |
| Breastfeed more frequently if baby is not getting enough breast milk | 43.8 | 46 | 44.6 |
| Mother of a <6 m old should not stop breastfeeding if the mother is ill | 95 | 93.7 | 96.9 |
| A baby should be breastfed till 24 m | 91.3 | 85.7 | 89.2 |
| A baby should first receive water or other liquid at 6 m | 48.8 | 47.6 | 43.8 |
| A baby should start semi-solid food at 6 m | 27.5 | 34.9 | 26.9 |
| A baby should start animal source foods at 6 m | 23.8 | 27 | 21.5 |
| A mother should not stop breastfeeding if the child is ill | 70 | 73 | 73.1 |
| >6 m old child with diarrhoea requires zinc | 96.3 | 96.8 | 94.6 |
| Knowledge score (scale 0 to 10) | (7.6 ± 1.5) | (7.8 ± 1.5) | (7.6 ± 1.5) |

[a] IYCF = infant and young child feeding

were significantly less satisfied with nutrition services, compared to those attending NGO facilities; and with increase in the number of services received, the satisfaction score increased.

For caregivers of young children, only the type of health facility was significantly associated with satisfaction with caregivers who used public facilities being less satisfied with nutrition services received than who used NGO facilities (Table 7).

## Discussion

After a decade of mainstreaming nutrition into the health services through NNS, our study was the first to assess the quality of nutrition service provision in urban Dhaka in terms of structural readiness, process of nutrition service delivery and outcome (client satisfaction). This work complements previous studies conducted in 2013–14 where researchers focused on nutrition services provided through public facilities in rural Bangladesh only [15]. Our study addresses that gap and provides evidence related to the quality of nutrition service provision in urban areas. Studies have demonstrated that the coverage of nutrition services were inequitable among urban residents [23]. Therefore, the insights from our study will provide an understanding of nutrition services delivery from supply side and contribute to formulation of future nutrition programs and policies for urban residents.

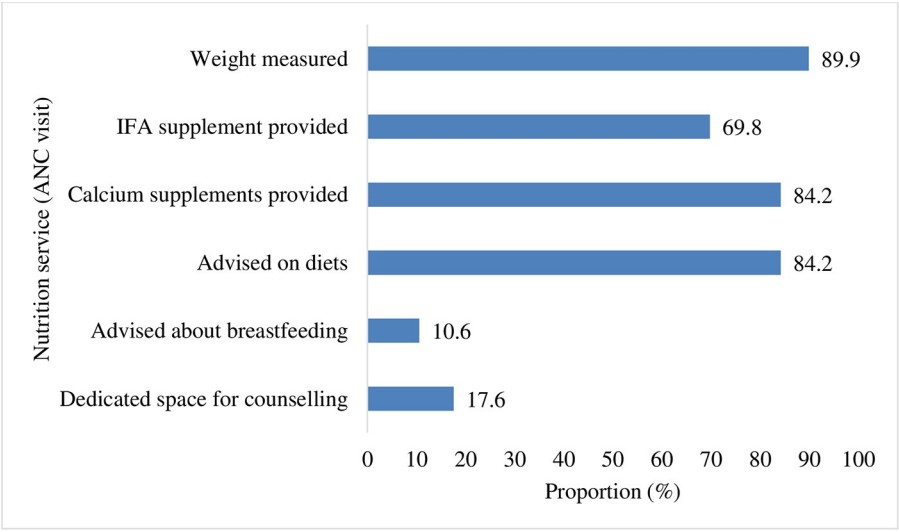

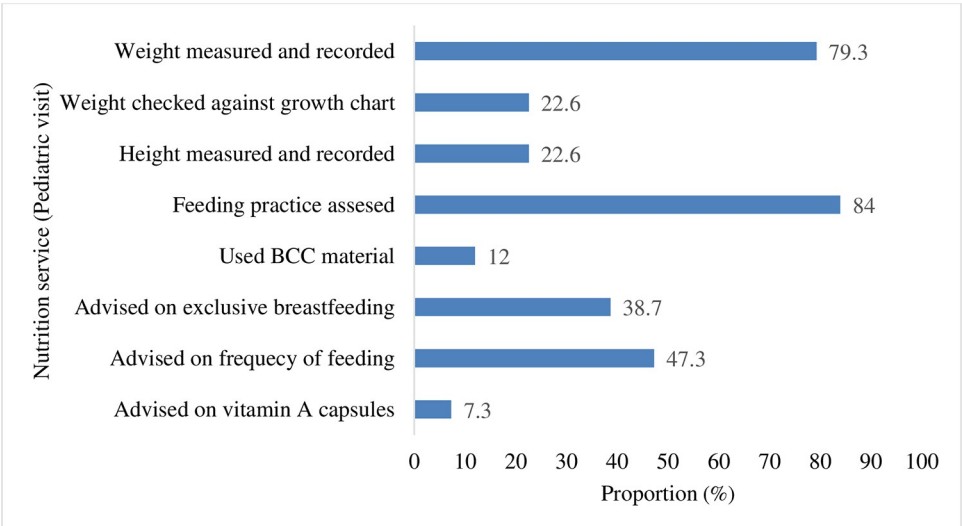

**Fig 1.** A. Proportion (%) of pregnant women who received nutrition services during ANC (n = 159); B: Proportion (%) of mothers/caregivers of young children who received nutrition services during paediatric visits (n = 150).

Among the different health facilities assessed, we found gaps in availability of equipment, guidelines and SBCC materials. Most facilities had weighing scales but many facilities did not have instrument for length and MUAC measurements, weight cards or growth cards that would allow monitoring of weight (pregnancy) and growth (children) over time. Many facilities did not have nutrition training manuals or SBCC materials available. In terms of nutrition training, there were gaps in basic NNS nutrition training among service providers (31–40% provider trained) although this proportion was higher than that reported from rural public health facilities (19%) in 2017 [15]. Healthcare providers had more knowledge about nutrition topics to include during counselling mothers of young children than pregnant women. Similar gaps in structural readiness of health facilities for nutrition service delivery have been observed in Bangladesh and other low- and middle-income countries (LMICs) [15, 24, 25]. Addressing

**Table 6. Satisfaction with nutrition services among of pregnant women and caregivers of young children.**

| Domain | Pregnant women | | Caregivers | |
|---|---|---|---|---|
| | n = 165 | | n = 162 | |
| | Mean ± SD | 95% CI | Mean ± SD | 95% CI |
| Waiting time | 3.4 ± 3.1 | (3.9, 2.9) | 3.6 ± 3.2 | (4.0, 3.0) |
| Waiting area | 3.9 ± 3.4 | (4.4, 3.4) | 3.9 ± 3.4 | (4.4, 3.4) |
| Quality of advice | 4.1 ± 3.6 | (4.6, 3.5) | 4.0 ± 3.5 | (4.6, 3.5) |
| Opportunity to ask questions | 4.3 ± 3.8 | (4.8, 3.7) | 4.3 ± 3.9 | (4.9, 3.7) |
| Respectful treatment | 4.3 ± 3.8 | (4.9, 3.7) | 4.2 ± 3.7 | (4.7, 3.6) |
| Intention to use the facility in the future | 4.4 ± 3.9 | (5, 3.8) | 4.3 ± 3.8 | (4.9, 3.7) |
| Recommend the facility to others | 4.3 ± 3.8 | (4.9, 3.7) | 4.3 ± 3.7 | (4.8, 3.7) |
| Overall satisfaction | 4.1 ± 3.6 | (4.7, 3.5) | 4.0 ± 3.6 | (4.6, 3.5) |

the gaps in nutrition service delivery should be addressed in a priority basis to improve the quality of services provided.

During observation of nutrition service delivery, we found that only 17.6% facilities had dedicated space for counselling and although most pregnant women received counselling about their own diet, only 10% women were counselled on breastfeeding. This is an important missed opportunity as previous studies have shown that counselling during pregnancy have significant positive impact on initiation and duration of breastfeeding [26, 27]. It is important to note that only half of the mothers received 4 ANC services which points to the need for improving health service utilization so that there are multiple opportunities to avail nutrition services. In this regard the determinants of existing inequities regarding ANC service use need to be addressed [28–30]. During paediatric visits we found that the quality of anthropometric measurement was low with very few cases of weight being recorded in a growth chart. Without good quality anthropometric measurement and its interpretation, it is unlikely that growth faltering can be detected, prevented and managed by the health care providers. Some of these gaps in implementation of nutrition services stem from lack of equipment and materials observed before [15]. However, it is also important to address service provider's attitude and knowledge related integrating nutrition services through other components of PHCs as previous studies showed the importance of service providers' attitude in implementing programs [31].

In terms of client satisfaction with nutrition services, satisfaction was quite high for both pregnant women and caregivers of young children. Similar high satisfaction scores for health and nutrition services have been reported in other studies despite gaps in facility readiness and quality of implementation [15, 32]. Researchers have observed that behaviour of the service providers play a greater role in client satisfaction rather than actual quality of service [33, 34]. Cleanliness of waiting area and waiting time received the lowest mean score similar to finding from other studies in Bangladesh and elsewhere [15, 24, 34]. Pregnant women's satisfaction was significantly greater with increasing number of nutrition services received. It is possible that higher number of nutrition services meant that service providers were spending more time with the clients which has been reported as an important determinant of satisfaction [35, 36]. For both pregnant women and caregivers of young children, satisfaction was significantly lower for public facilities compared other facilities. Researchers have reported increase in caseloads in public facilities leading to long waiting time and therefore, less satisfaction with the services received [34–36].

Urban areas contain 38% of the population of Bangladesh [37] and about 24% of urban population are the urban poor [38]. Given the high prevalence of malnutrition among mothers

**Table 7. Factors associated with satisfaction score related to health service utilization among pregnant women and caregivers of young children.**

| Variables | Unadjusted model | | | Adjusted model | | |
|---|---|---|---|---|---|---|
| | Coefficient | 95% CI | p-value | Coefficient | 95% CI | p-value |
| **Pregnant women (n = 165)** | | | | | | |
| *Age(years)* | | | | | | |
| 13–19 | 0.204 | (0.003, 0.404) | 0.046 | 0.182 | (0.004, 0.361) | 0.045 |
| 20–24 | 0.089 | (-0.090, 0.269) | 0.327 | 0.089 | (-0.069, 0.247) | 0.269 |
| 25–30 | 0.231 | (0.048, 0.414) | 0.014 | 0.258 | (0.093, 0.422) | 0.002 |
| <30 | Ref | - | - | Ref | - | - |
| *Gestational age* | | | | | | |
| 1st trimester | 0.078 | (-0.092, 0.249) | 0.365 | 0.058 | (-0.096, 0.212) | 0.458 |
| 2nd trimester | 0.056 | (-0.088, 0.200) | 0.445 | 0.014 | (-0.114, 0.141) | 0.834 |
| 3rd trimester | Ref | - | - | Ref | - | - |
| *Type of health facility* | | | | | | |
| Government | -0.299 | (-0.444, -0.154) | <0.001 | -0.295 | (-0.430, -0.161) | <0.001 |
| Private | -0.117 | (-0.262, 0.028) | 0.113 | -0.081 | (-0.220, 0.059) | 0.258 |
| NGO | Ref | - | - | Ref | - | - |
| *Number of nutrition services received* | | | | | | |
| Below group mean | -0.219 | (-0.338, -0.100) | <0.001 | -0.194 | (-0.305, -0.084) | 0.001 |
| Above group mean | Ref | - | - | Ref | - | - |
| **Caregivers (n = 162)** | | | | | | |
| *Child age (month)* | | | | | | |
| 0 to 6 | -0.022 | (-0.147, 0.103) | 0.726 | -0.029 | (-0.154, 0.096) | 0.650 |
| 7 to 24 | Ref | - | - | Ref | - | - |
| *Reason to visit facility* | | | | | | |
| Sick child visit | 0.191 | (-0.209, 0.591) | 0.347 | 0.176 | (-0.222, 0.575) | 0.386 |
| Immunization | 0.214 | (-0.218, 0.647) | 0.329 | 0.205 | (-0.220, 0.630) | 0.344 |
| Others | Ref | - | - | Ref | - | - |
| *Type of health facility* | | | | | | |
| Government | -0.166 | (-0.315, -0.018) | 0.029 | -0.170 | (-0.319, -0.022) | 0.025 |
| Private | -0.087 | (-0.237, 0.064) | 0.259 | -0.095 | (-0.247, 0.058) | 0.223 |
| NGO | Ref | - | - | Ref | - | - |
| *Number of nutrition services received* | | | | | | |
| Below group mean | -0.035 | (-0.174, 0.105) | 0.625 | -0.001 | (-0.140, 0.137) | 0.985 |
| Above group mean | Ref | - | - | Ref | - | - |

and young children, especially the urban poor [39], it is imperative that nutrition service delivery is strengthened in urban areas. Bangladesh already has many strategies and guidelines that support the implementation nutrition services [40]. However, the gaps in service provision needs to be addressed urgently if Bangladesh is to achieve sustainable development goal 2 related to malnutrition reduction. Currently for children, mothers avail health services for immunization and when a child is sick. For nutrition services such as anthropometric measurement and counselling to work efficiently it is important to design well child visits within the health services which may require rethinking how PHC is currently delivered.

This study had a few limitations. We did not apply probability proportional to size (PPS) method for sampling of the health facilities and thus our results are not representative of national data. Moreover, we excluded health facilities serving less than 15 clients per day. This kind of sampling may have overestimated the service quality if health facilities with greater patient load are better equipped to deliver nutrition services than the smaller ones. The study

was conducted in Dhaka, the largest city in Bangladesh and may not reflect the realities of smaller cities. Direct observation of interactions between clients and health service provider to understand the process of nutrition service delivery may have changed the way service was delivered. Our study has several strengths. The study was conducted after a decade of mainstreaming nutrition through PHC therefore, the data was a good reflection of how NNS have been implemented through urban health facilities. We assessed the quality of nutrition service delivery in primary care facilities implemented by a diverse type of organizations (public, private, NGO) which reflects the realities of urban cities.

## Conclusion

In this study we assessed the gaps in facility readiness, quality of nutrition service delivery and client satisfaction with nutrition service provision in urban health facilities. The impact of the gaps in availability of equipment and guidelines, and lack of nutrition training of health personnel meant that health personnel were not adequately prepared to talk about nutrition with the mothers. Anthropometric measurement is an important part of nutrition specific services as this can provide decision support for health personnel and gaps in the quality of these measurements can hamper early detection and management of malnutrition among pregnant women and children. It is important that with the experience of a decade of mainstreaming nutrition through health services, a national consultative process is initiated under the leadership of Ministry of Health and Family Welfare to design opportunities of good quality nutrition counselling within existing health services. Finally, although user satisfaction with the nutrition services were quite high, the gaps in terms of quality of the waiting area and waiting time, especially for public facilities, need to be improved so that mothers can avail nutrition services in comfort and are motivated to use the facilities. Urban areas cater to almost a third of Bangladeshi population and the level of malnutrition among the urban poor (specially children) are high. Improving the quality of nutrition services could help address the needs of the urban residents and contribute to achieving Sustainable Development Goal 2 related to ending malnutrition for Bangladesh.

## Supporting information

**S1 Table. List of health facilities.**
(PDF)

**S2 Table. Scoring of knowledge questions about nutrition counselling among health service providers.**
(PDF)

**S1 Dataset.**
(RAR)

## Acknowledgments

We acknowledge the leadership and technical guidance of the Institute of Public Health Nutrition of the Ministry of Health and Family Welfare. We are grateful to the media firm who helped us with conducting the media analysis. We also thank the retail store managers and owners for their time and cooperation. We acknowledge the contribution of Dr Shafiqul Alam Sarker for reviewing the study proposal. icddr,b is also grateful to the Governments of Bangladesh, Canada, Sweden, and the UK for providing core/unrestricted support.

## Author Contributions

**Conceptualization:** Syeda Mahsina Akter, Santhia Ireen, Sabrina Rasheed.

**Data curation:** Faugia Islam Anne.

**Formal analysis:** Faugia Islam Anne.

**Funding acquisition:** Sabrina Rasheed.

**Investigation:** Sabrina Rasheed.

**Methodology:** Faugia Islam Anne, Sabrina Rasheed.

**Resources:** Jessica Escobar-DeMarco, Deborah Ash.

**Supervision:** Jessica Escobar-DeMarco, Sabrina Rasheed.

**Validation:** Kristen Kappos, Sabrina Rasheed.

**Writing – original draft:** Faugia Islam Anne, Sabrina Rasheed.

**Writing – review & editing:** Faugia Islam Anne, Sifat Parveen Sheikh, Santhia Ireen, Jessica Escobar-DeMarco, Kristen Kappos, Deborah Ash, Sabrina Rasheed.

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
