## [Decision Letter · Decision Letter 0]

18 Aug 2022

PONE-D-22-18876Quality of nutrition services in primary health care facilities of Dhaka city: state of nutrition mainstreaming in urban BangladeshPLOS ONE

Dear Dr. Rasheed,

Thank you for submitting your manuscript to PLOS ONE. After careful consideration, we feel that it has merit but does not fully meet PLOS ONE’s publication criteria as it currently stands. Therefore, we invite you to submit a revised version of the manuscript that addresses the points raised during the review process.

We look forward to receiving your revised manuscript.

Kind regards,

Haribondhu Sarma, MSS, PhD

Academic Editor

PLOS ONE

Journal Requirements:

Additional comments from Academic Editor:

Comments to the author(s)

The "Quality of nutrition services in primary health care facilities of Dhaka City: state of nutrition mainstreaming in urban Bangladesh’ is an important and interesting manuscript that identified system-level gaps in nutrition service delivery. The study uses cutting-edge methods to analyse nutrition service quality in urban Bangladesh. Peer reviewers also found this paper interesting and suggested some modifications. I also have the following comments for better clarity of this paper:

Title:

Authors may remove "primary" from the title as the paper considered both primary and secondary healthcare facilities (I see in table 1 that 20 secondary healthcare facilities are included in the analysis).

Abstract:

Please define ANC in the abstract. Please add findings with the sub-heading "Structural readiness", as they have for "process" and "outcome". Or take out all.

Introduction:

The introduction section does not adequately explain the existing urban nutrition services in Bangladesh. Readers may benefit from this description, please describe urban nutrition services in Bangladesh in line with the current Operations Plan for NNS.

Methods:

Please explain how the WHO/UNICEF NetCode protocol works for assessing health care facilities and how far the authors have used this protocol in this study.

Please add a functional definition of "quality nutrition services" as considered for this study.

Report how many service providers and mothers did not consent to being interviewed (non-response rate?), and how they were considered in the analysis, any statistical implications.

Table 2 does not clarify the indicators adequately. In particular, what are the process and outcome indicators considered for this assessment?

Line 128: Please add a comma (,) after the guidelines. For supplements, please specify whether they are capsules or tablets, (e.g., Vitamin A capsule; Iron, Folic Acid (IFA) tablet, Calcium tablet)?

Line 134-137: Please clarify what are the items of the 10 questions for calculating the knowledge score. How correct answers were assessed? Were they multiple choice questions or open-ended? I propose that the authors include a supplementary table to further clarify each of the knowledge items. I am also having trouble understanding how 10-questions’ knowledge scores ranged between 0-20, given that each correct question scored 1.

Line 153-54: Please specify the independent variables (e.g., type of health facility?) that are considered for multivariable modelling.

Discussion:

In the limitation, please acknowledge that the study did not fully comply with the WHO/UNICEF NetCode protocol and explain whether it affected the integrity of the study, if not how?

Conclusion:

Line 331: Use the acronym (MoHFW) as already defined earlier.

References:

The references of the paper do not comply with the PLoS One recommended reference style. Please revise and format all references thoroughly.

Figures:

Resolutions for both the figures (1A and 1B) need to be improved further. Please check

PLoS One figure submission guidelines.

Overall, the paper needs careful proof editing for fixing some typos throughout the paper.

Reviewers' comments:

Reviewer's Responses to Questions

**Comments to the Author**

1. Is the manuscript technically sound, and do the data support the conclusions?

Reviewer #1: Yes

Reviewer #2: Yes

2. Has the statistical analysis been performed appropriately and rigorously? 

Reviewer #1: Yes

Reviewer #2: Yes

3. Have the authors made all data underlying the findings in their manuscript fully available?

Reviewer #1: Yes

Reviewer #2: Yes

4. Is the manuscript presented in an intelligible fashion and written in standard English?

Reviewer #1: Yes

Reviewer #2: Yes

5. Review Comments to the Author

Reviewer #1: In this manuscript, authors have attempted to explore the quality of nutrition service provision at public primary health care facilities in urban Dhaka which is very timely research for Bangladesh. The findings of this study will guide the policy makers and implementer to plan interventions to improve the health services in the urban context. This is a very well written manuscript with sound statistical analysis.

Minor comments:

Line 86: ‘From the list we identified health facilities that provided ANC and 87 postnatal care (PNC), delivery services, and pediatric care…’. It would be great asset for the readers if you add the list of health facilities as a supplementary file.

Conclusion: I would suggest the author to provide specific, bold and focused 3-4 implications of the findings which could help the reader to have an idea about way forward and also to suggest some further research areas.

Reviewer #2: This research article is an effort to fill gap in the knowledge base of quality of nutrition services provided by primary and secondary healthcare providers in Dhaka city and reveals the state of nutrition services in urban setting in Bangladesh. I would recommend to accept the article for publishing, with subject to the following minor revisions:

1. In the title and abstract, you described that this research is about scrutinizing the primary healthcare services, however, you have taken samples from the secondary healthcare providers also. This should be addressed throughout the manuscript consistently.

2. In methodology part, you said about multivariate analysis, but did not say particularly which sort of multivariate analysis was performed. For example, I guess there were two different models (possibly linear regression with factor variables, not mentioned explicitly) for two different type of respondents, as they have different set of independent variables. These need to be elaborated more precisely.

3. In describing the multivariate analyses, the authors mentioned about Table 8, which is non-existent.

4. Regarding the tools, you said that minor adjustment in the context of Bangladesh has been made. However, the source tools were not described explicitly, let alone mentioning about the adjustment in specific.

5. In the introduction of abstract (line 26) there is mention of only public facilities, whereas, in results section (line 42) there are mentions of public, NGO and private service providers.

6. In discussion section, while describing the structural readiness of the providers, you described the percentage of facilities or receivers (line 273-276) having the services or about receiving the services, respectively. I would suggest to mention the figures in reverse (mentioning the % of negative side), because only then the bigger gap with bigger number would be more visible to the reader or policy makers.

7. In discussion (line 306), you mentioned about Sustainable Development Goal 2, but did not mentioned about the goal in few words. I think, for general readers, it should be mentioned in text. Further, the goal in general is about ending hunger, achieving food security and improving nutrition. So, you should mention specifically which sub-goal you intended to mention.

8. One of the missed opportunity in the discussion section is that there are absence of illustrating the outcome in the light of multivariate models. Rather, bi-variate results were mainly explained. For example, the authors could mention about that the pregnant mothers are sensitive on their satisfaction level (significantly) if under age 19 or over age 25. Further, satisfactions are significantly reducing while the recipient are receiving them from the Government facilities (p<0.001). These results and their policy implication should be reflected in the discussion section.

9. Finally, I would prefer the graphs with data value labels for the bars/columns.

6. PLOS authors have the option to publish the peer review history of their article (what does this mean?). If published, this will include your full peer review and any attached files.

Reviewer #1: No

Reviewer #2: **Yes: **Kabir Ahmad

---

## [Author Response · Author response to Decision Letter 0]

31 Oct 2022

Rebuttal Letter

Editor’s comments:

Author’s responses: Thanks. We have addressed according to PLOS One style requirements. 

Author’s responses: We addressed this in the method section of manuscript (line no 175). 

Author’s responses: Thanks. We shared the data set . 

Author’s responses: We shared the data set. 

Author’s responses: We shared the data set .

We note that you have included the phrase “data not shown” in your manuscript. Unfortunately, this does not meet our data sharing requirements. PLOS does not permit references to inaccessible data. We require that authors provide all relevant data within the paper, Supporting Information files, or in an acceptable, public repository. Please add a citation to support this phrase or upload the data that corresponds with these findings to a stable repository (such as Figshare or Dryad) and provide and URLs, DOIs, or accession numbers that may be used to access these data. Or, if the data are not a core part of the research being presented in your study, we ask that you remove the phrase that refers to these data.

Author’s responses: We have omitted “data not shown” phrase and we shared the data set

Additional comments from Academic Editor:

Comments to the author(s)

The "Quality of nutrition services in primary health care facilities of Dhaka City: state of nutrition mainstreaming in urban Bangladesh’ is an important and interesting manuscript that identified system-level gaps in nutrition service delivery. The study uses cutting-edge methods to analyze nutrition service quality in urban Bangladesh. Peer reviewers also found this paper interesting and suggested some modifications. I also have the following comments for better clarity of this paper:

a. Title:

Authors may remove "primary" from the title as the paper considered both primary and secondary healthcare facilities (I see in table 1 that 20 secondary healthcare facilities are included in the analysis).

 Author’s responses: We have removed the word “primary” from title. 

b. Abstract:

Please define ANC in the abstract. Please add findings with the sub-heading "Structural readiness", as they have for "process" and "outcome". Or take out all.

Author’s responses: We defined the word ANC in abstract (line no 31). We removed subheading in abstract. (line no 37 and 40). 

c. Introduction:

The introduction section does not adequately explain the existing urban nutrition services in Bangladesh. Readers may benefit from this description, please describe urban nutrition services in Bangladesh in line with the current Operations Plan for NNS.

Author’s responses: We have described urban nutrition services in Bangladesh in line with the current Operations Plan for NNS in introduction section (line no 65 to 70).

d. Methods:

Please explain how the WHO/UNICEF NetCode protocol works for assessing health care facilities and how far the authors have used this protocol in this study.

Author’s responses: We addressed in the method section (please see line no 99-104).

e. Please add a functional definition of "quality nutrition services" as considered for this study.

Author’s responses: Thanks. Please check line no 133-135. 

f. Report how many service providers and mothers did not consent to being interviewed (non-response rate?), and how they were considered in the analysis, any statistical implications.

Author’s responses: We reported this in the method section (line no 114-118).

g. Table 2 does not clarify the indicators adequately. In particular, what are the process and outcome indicators considered for this assessment?

Author’s responses: We added details of process and outcome indicators in the table 2 (line no 137).

h. Line 128: Please add a comma (,) after the guidelines. For supplements, please specify whether they are capsules or tablets, (e.g., Vitamin A capsule; Iron, Folic Acid (IFA) tablet, Calcium tablet)?

Author’s responses: Addressed (line no 140).

i. Line 134-137: Please clarify what are the items of the 10 questions for calculating the knowledge score. How correct answers were assessed? Were they multiple choice questions or open-ended? I propose that the authors include a supplementary table to further clarify each of the knowledge items. I am also having trouble understanding how 10-questions’ knowledge scores ranged between 0-20, given that each correct question scored 1.

Author’s responses: Thank you for your useful suggestion. We have added supplementary table for knowledge items (S2 table). We corrected the score range. 

j. Line 153-54: Please specify the independent variables (e.g., type of health facility?) that are considered for multivariable modelling.

Author’s responses: Addressed (line no 168-170). 

k. Discussion:

In the limitation, please acknowledge that the study did not fully comply with the WHO/UNICEF NetCode protocol and explain whether it affected the integrity of the study, if not how?

Author’s responses: Thank you. We have addressed in the limitation section (line no 329-331). 

l. Conclusion:

Line 331: Use the acronym (MoHFW) as already defined earlier.

Author’s responses: We addressed accordingly.

m. References:

The references of the paper do not comply with the PLoS One recommended reference style. Please revise and format all references thoroughly.

Author’s responses: We have checked and addressed accordingly. 

n. Figures:

Resolutions for both the figures (1A and 1B) need to be improved further. Please check

PLoS One figure submission guidelines.

Author’s responses: Thank you. We have addressed accordingly. 

o. Overall, the paper needs careful proof editing for fixing some typos throughout the paper.

Author’s responses: We have checked thoroughly and addressed accordingly.

5. Review Comments to the Author

Reviewer #1: In this manuscript, authors have attempted to explore the quality of nutrition service provision at public primary health care facilities in urban Dhaka which is very timely research for Bangladesh. The findings of this study will guide the policy makers and implementer to plan interventions to improve the health services in the urban context. This is a very well written manuscript with sound statistical analysis.

A. Minor comments:

Line 86: ‘From the list we identified health facilities that provided ANC and 87 postnatal care (PNC), delivery services, and pediatric care…’. It would be great asset for the readers if you add the list of health facilities as a supplementary file.

Author’s responses: We shared list of health facilities 

B. Conclusion: I would suggest the author to provide specific, bold and focused 3-4 implications of the findings which could help the reader to have an idea about way forward and also to suggest some further research areas.

Author’s responses: Thank you for your useful suggestion. We addressed this (line no 346-348).

Reviewer #2: This research article is an effort to fill gap in the knowledge base of quality of nutrition services provided by primary and secondary healthcare providers in Dhaka city and reveals the state of nutrition services in urban setting in Bangladesh. I would recommend to accept the article for publishing, with subject to the following minor revisions:

1. In the title and abstract, you described that this research is about scrutinizing the primary healthcare services, however, you have taken samples from the secondary healthcare providers also. This should be addressed throughout the manuscript consistently.

Author response: Thank you. Though we used sample from secondary health facilities but we mainly focused on primary health care service delivery in those secondary health facilities. 

2. In methodology part, you said about multivariate analysis, but did not say particularly which sort of multivariate analysis was performed. For example, I guess there were two different models (possibly linear regression with factor variables, not mentioned explicitly) for two different type of respondents, as they have different set of independent variables. These need to be elaborated more precisely.

Author’s responses: We addressed in method section (line no 169-171). 

3. In describing the multivariate analyses, the authors mentioned about Table 8, which is non-existent.

Author’s responses: We addressed.

4. Regarding the tools, you said that minor adjustment in the context of Bangladesh has been made. However, the source tools were not described explicitly, let alone mentioning about the adjustment in specific.

Author’s responses: Spring tool was used as our source tool. Reference has been given in line no 128. 

5. In the introduction of abstract (line 26) there is mention of only public facilities, whereas, in results section (line 42) there are mentions of public, NGO and private service providers.

Author’s response: We addressed accordingly. 

6. In discussion section, while describing the structural readiness of the providers, you described the percentage of facilities or receivers (line 273-276) having the services or about receiving the services, respectively. I would suggest to mention the figures in reverse (mentioning the % of negative side), because only then the bigger gap with bigger number would be more visible to the reader or policy makers.

Author’s responses: Thank you for your feedback. We have addressed as per your feedback (line no 282-284). 

7. In discussion (line 306), you mentioned about Sustainable Development Goal 2, but did not mentioned about the goal in few words. I think, for general readers, it should be mentioned in text. Further, the goal in general is about ending hunger, achieving food security and improving nutrition. So, you should mention specifically which sub-goal you intended to mention.

Author’s responses: We have addressed accordingly (line no 361). 

8. One of the missed opportunity in the discussion section is that there is absence of illustrating the outcome in the light of multivariate models. Rather, bi-variate results were mainly explained. For example, the authors could mention about that the pregnant mothers are sensitive on their satisfaction level (significantly) if under age 19 or over age 25. Further, satisfactions are significantly reducing while the recipient are receiving them from the Government facilities (p<0.001). These results and their policy implication should be reflected in the discussion section.

Author’s responses: Thank you for your suggestion. We have addressed this in the discussion section (line no 314-317). 

9. Finally, I would prefer the graphs with data value labels for the bars/columns.

Author’s responses: Data value labels has been added. 

6. PLOS authors have the option to publish the peer review history of their article (what does this mean?). If published, this will include your full peer review and any attached files.

Do you want your identity to be public for this peer review? For information about this choice, including consent withdrawal, please see our Privacy Policy.

Reviewer #1: No

Reviewer #2: Yes: Kabir Ahmad

---

## [Editor Report · Decision Letter 1]

21 Nov 2022

Quality of nutrition services in health care facilities of Dhaka city: state of nutrition mainstreaming in urban Bangladesh

PONE-D-22-18876R1

Dear Dr. Rasheed,

We’re pleased to inform you that your manuscript has been judged scientifically suitable for publication and will be formally accepted for publication once it meets all outstanding technical requirements.

Kind regards,

Haribondhu Sarma

Academic Editor

PLOS ONE
---

## [Editor Report · Acceptance letter]

5 Dec 2022

PONE-D-22-18876R1 

Quality of nutrition services in primary health care facilities of Dhaka city: state of nutrition mainstreaming in urban Bangladesh 

Dear Dr. Rasheed:

I'm pleased to inform you that your manuscript has been deemed suitable for publication in PLOS ONE. Congratulations! Your manuscript is now with our production department. 

Kind regards, 

on behalf of

Dr. Haribondhu Sarma 

Academic Editor

PLOS ONE